# Why We Should Study Checklists: Reflections on an Overlooked Idea

Hajo A. Reijers[1][0000-0001-9634-5852] and Henrik
Leopold[2][0000-0003-4862-1829]

[1] Utrecht University, Utrecht, The Netherlands
`h.a.reijers@uu.nl`
[2] Kühne Logistics University, Hamburg, Germany
`henrik.leopold@klu.org`

**Abstract.** Checklists are among the most widely used instruments for guiding work, yet they have received little attention in Business Process Management (BPM) research. In 2017, we published a paper that proposed to view checklists as informational artifacts and called for a "science of checklists". The paper proposed a conceptualization of checklists in terms of seven properties, analyzed twenty-one recurring problems with their use, and argued that a design-oriented, informational approach could be used to overcome many of these problems. The paper was published and has since been cited about thirty times. However, the research agenda it set out was not taken up and a grant proposal intended to pursue it was rejected at an early stage. In this paper, we revisit this idea as a case of an overlooked research direction. We reconstruct its argument, discuss why a published and cited idea failed to gain traction, and argue that recent developments in process mining, large language models, and AI agents make the agenda more relevant than before. Our wider aim is to ask what it means that the BPM community has paid so little attention to the artifact that arguably guides work in practice more than any other.

**Keywords:** Business process management · Checklists · Informational artifacts · Research agendas · Forgotten topics.

## 1 Introduction

Checklists are ubiquitous. People use them to do their shopping, pilots use them before take-off, and surgical teams use them to keep patients safe. In each case, the checklist serves a simple but important purpose: It guides people through the steps of a work process and helps them verify that nothing has been forgotten. Few instruments for guiding work are as widely used. It is therefore striking how little attention checklists have received in the field that concerns itself with how work processes can be designed and carried out, namely Business Process Management (BPM).

In 2017, we tried to address this gap. We published a paper in which we argued that checklists deserve to be studied as informational artifacts, much the same

way as process models or other representations of work [9]. The paper proposed a way to conceptualize checklists, analyzed the problems that occur with their use, and called for a "science of checklists" that the BPM and Information Systems (IS) communities would be well placed to develop.

What happened to this idea is the reason for the present paper. The paper was first rejected at ECIS in 2016 and then accepted and published at HICSS in 2017. Since then, it has been cited about thirty times. These citations confirm that the paper was read. However, on closer inspection, the citing papers refer to the idea rather than build on it. No substantial follow-up work has taken up the research agenda we proposed. The grant proposal that was intended to pursue this agenda was rejected at the pre-proposal stage. In other words, the idea was registered, but not taken up. Therefore, we consider it an overlooked idea.

It is worth asking why this should concern anyone beyond ourselves. We believe it should. Consider the technologies that our field has studied most intensively as means of coordinating and guiding work. The Business Process Management System (BPMS) has been a central object of study for decades. More recently, blockchain technology has attracted intense interest as a way to execute and coordinate processes across organizational boundaries, and Robotic Process Automation (RPA) has done the same for routine office work. Each of these has generated substantial research literature. Yet none of them is anywhere near universally adopted in practice: the BPMS remains largely confined to particular organizations and particular kinds of process; blockchain-based process execution has seen little uptake outside research; and even RPA, the most successful of the three commercially, touches only a fraction of the work that people actually carry out.

The checklist is the mirror image. It has drawn almost no attention in our field, while it is, by any reasonable measure, the most widely used instrument in the world for guiding people through their work, from the operating theater to the cockpit to the kitchen. In other words, the BPM field has devoted attention to a succession of sophisticated technologies that comparatively few people use to guide their work, while overlooking a simple tool that nearly everyone does. What does this say about our priorities? We suspect that it is this: that BPM research may be driven more by intellectual puzzles and sophisticated technology than by impact on real-world practice. Seen in this way, the fate of our paper is a small symptom of a larger blind spot, which makes it worth revisiting here.

The case is, if anything, more pressing today. As the field turns its attention to AI agents as a new means of carrying out work, we will argue that this newest technology depends on the very artifact it has overlooked.

The remainder of the paper is organized as follows. In Section 2, we will reconstruct the original idea. Then, we will reflect on how it was received in Section 3 and the factors behind that reception in Section 4. Section 5 looks forward to reviving work on checklits, after which we conclude the paper in Section 6.

## 2   The Overlooked Contribution

### 2.1   The Idea: Checklists as Informational Artifacts

The starting point of our 2017 checklist paper [9] was a paradox. On the one hand, checklists can be highly effective. The best-known example is the Surgical Safety Checklist of the World Health Organization, whose introduction has been associated with a halving of mortality in hospitals that adopted it [7]. Similar gains have been reported in aviation and other high-risk settings. On the other hand, checklists are subject to a range of well-documented problems and their adoption is uneven: A checklist that is highly effective in one organization may be resisted, used incorrectly, or abandoned in another. Given their demonstrated value, this gap between what checklists can achieve and how they are used in practice is remarkable.

In the seed paper, we collected seventy-one issues from the literature and grouped them into twenty-one recurring types. The most frequently reported is that a checklist is not sensitive to the context or the case at hand. Others include non-compliance, "checklist fatigue" when too many lists compete for attention, an over-reliance on human judgment, an inability to deal with exceptions, and the false impression that work has been carried out correctly once all items have been checked. A further group of problems, concerning how checklists are kept up to date and managed as they accumulate, had received almost no attention. Our fundamental analysis is that these are not incidental difficulties that, for example, can be resolved by better training, Rather, they appear to be rooted in the design and the management of the artifact.

We observed that the literature offered no systematic account of why these problems arise or how they might be addressed in a principled way, despite decades of checklist use in many domains. Existing solutions shared two limitations. First, they were domain-specific: Aviation experts worked on aviation checklists, medical experts on medical checklists, and the lessons did not travel between them. Second, they addressed the surface of the checklist, such as its organizational uptake, rather than any fundamental property of the checklist itself. The result was a body of practical experience but little cumulative knowledge: no shared vocabulary for what a checklist is, no account of which features of a checklist give rise to which problems, and as a result of that no basis on which a solution found in one domain could be transferred to another.

For these reasons, our central proposal was to regard the checklist as an *informational artifact*, that is, as a conceptual model that both represents a work routine and prescribes the actions and decisions within it. Seen in this way, the checklist belongs to the same family as process models, grammars of actions, organizational routines, and state machines. Then it also becomes accessible to the methods that BPM and IS research applies to such artifacts. From the same structured literature survey, we derived a conceptualization of checklists in terms of seven properties: five that describe the checklist as a whole (representation, prescriptiveness, scope, abstraction, and audience) and two that describe its items (type and behavioral relation). This step allowed us to relate the recurring

problems to specific properties and to argue about which features of a checklist cause which of its problems.

## 2.2   The Associated Grant Proposal

The seed paper was intended to be the start of a larger programme. Shortly after its publication, the first author submitted that programme to the Dutch Research Council (NWO) as a VICI proposal titled "Checklist Science". The VICI is the most senior grant in the NWO Talent Programme and is intended precisely to let an established researcher build a new line of work over five years, so the proposal was the natural attempt to grow the seed into a research line.

The proposal took the information-processing view of the seed paper and turned it into four concrete lines of development: (1) a generic language for checklists; (2) technology for automatically updating a checklist from available data, so that it becomes a "living" document that imposes less burden on its users; (3) a technique for mining the data that the use of checklists generates; and (4) a mechanism for configuring checklists to the case at hand. Each line was meant to address one of the persistent checklist problems identified in the seed paper. The work was to be carried out together with the SURPASS Support Group and the Academic Medical Center in Amsterdam, using their digital-checklist facilities to conduct pilots in hospitals that already use the SURPASS surgical checklist. The explicit aim of this collaboration was to improve existing checklists and extend their use to new domains.

## 3   The Record

It is useful to reflect on what happened to this idea because the pattern is part of our argument. The seed paper was first submitted as work-in-progress to ECIS in 2016, under the title "Fundaments for a Science of Checklists", where it was rejected. We revised it and it was accepted at HICSS as a full paper in 2017, under the title "Towards a Science of Checklists", where it was published. Since then, it has been cited about thirty times.

Obviously, being cited is not the same as being built upon. We followed the citing literature over the years and noticed that the paper is mostly cited in passing, typically to acknowledge that checklists matter or to borrow one of the problems we cataloged, and not as a basis for new work. For example, a recent narrative review of checklists and error-reporting systems in hospital care lists our article among its references without engaging with its argument [5]. We have not seen any study that adopts our seven properties, our problem categories, or our solution strategies as the foundation for a new artifact or method. The one neighboring effort in BPM, a tool that turns process models into paper "Process Checklists" [2], rests on a notion of the checklist [3] that predates our paper. So, it runs parallel to our idea rather than building on it. In other words, the research agenda that the paper sets out has not been taken up.

The funding history points in the same direction. The first author submitted the agenda as a VICI proposal to NWO. Of the 19 VICI pre-proposals in the relevant domain that year, 13 received a positive recommendation to proceed to a full application but this one was among the six that did not. The decision was made at the pre-proposal stage, which is the point where the breadth and promise of an idea are judged, rather than the detail of its execution.

What is striking is how closely the two evaluations, a year apart, agree. In both cases, the reviewers found the work clearly written and the topic relevant, yet in both cases they could not see where the scientific challenge lay and judged the contribution to be vague. The ECIS reviewers were, moreover, divided over whether the work belonged in a BPM track at all. One remarked that not every two fields deserve to be connected, while the most expert reviewer recommended that we keep the description of checklists but remove the section that linked them to BPM and information technology, which was the very connection we were trying to make. The NWO committee, while rating the applicant highly, asked what the scientific challenges of the proposal were and suggested that a more original approach would have been appropriate.

## 4    Diagnosis: Why Was the Idea Overlooked?

Why would an idea that was published and cited, is connected to a real-world problem of high relevance, and could be followed up with the methods and techniques of the BPM and IS communities fail to gain traction? We can think of several explanations.

*Explanation 1: Citation is not the same as uptake.* A paper can be cited as a convenient reference, for instance, to acknowledge that checklists matter, without prompting anyone to do the work it proposes. Some thirty citations that mention the idea but do not extend it are, on this view, evidence of overlooking rather than against it. The single adjacent BPM effort developed its own notion of the checklist independently, which shows that even researchers working on checklists in BPM did not take our framework as a starting point.

*Explanation 2: The checklist appears trivial.* We anticipated this perceived lack of complexity in the paper itself, where we noted that checklists are easily seen as trivial artifacts that can be left to the professionals of each application domain. The evaluations bear this out: the recurring question of where the scientific challenge lay treats the checklist as a place where, almost by definition, no deep scientific problem is expected to reside.

*Explanation 3: The idea did not fit the centre of gravity of the field.* In the relevant period, BPM and IS research concentrated on process models, while event-data analysis and process mining were still upcoming topics, and the checklist matched neither. This is what the ECIS reviews registered when they questioned whether the work belonged in BPM at all and advised keeping the description of checklists

but dropping their link to BPM and information technology. The very connection we saw as the direction of a solution did not fit the mindset of the community.

*Explanation 4: Agenda-setting ideas are filtered out early.* A broad agenda-setting proposal trades details for reach. The NWO decision illustrates the point: the committee rated the applicant highly but found the plan vague and the state of the art under-described, so stopped the proposal at the pre-proposal stage. These are the weaknesses that a low-information filter is designed to detect, so a proposal that aims for breadth may be the kind most exposed to them.

## 5   The Case for Revival

We should acknowledge that some of the criticism that we received was fair. The VICI pre-proposal may have appeared vague about the precise scientific challenges it would tackle, while the state of the art could have been described in more detail. The earlier ECIS version lacked an explicit methodology and clear research questions. Moreover, it did not distinguish the checklist cleanly from neighboring notions such as the to-do list. These are weaknesses that could be addressed in part in the HICSS paper, but none of these, in our opinion, diminish the value of the idea itself.

In addition, we believe that the case for taking on a research agenda is more attractive now than it was in 2017. First, the problems persist. Recent work continues to document poor and incomplete compliance, as well as a "tick-box" culture in which recording completion can displace the work the checklist is meant to support [6]. There is still an absence of systematic and principled methods for designing checklists in the first place [8]. Also, reviews of their effectiveness remain decidedly mixed [1].

Second, the means of addressing the problems we identified have improved. Previously, the "smart machine" we described was largely fictitious. Today, three developments make it concrete: 1) process mining has become an established way of learning from the data that work processes generate, 2) Large Language Models (LLMs) make it possible to generate, adapt, and interpret checklist items, including the interrogative and branched item types we identified, and 3) AI agents supply an executor that can carry, maintain, and act on a checklist rather than leaving each item to be ticked by hand. Taken together, these are the components of the "living" checklist we once imagined. The lines of the original proposal can now be stated as a concrete, viable, and relevant research agenda:

1. A generic language for representing checklists, precise enough to express the seven properties and to support analysis;
2. "Living" checklists that update themselves from available data, so that an item's status need not be maintained by hand;
3. Mining of the data generated by checklist use, to anticipate which items are likely to become relevant or critical;
4. Configuration and variant management, so that a checklist can be tailored to a case and a family of checklists managed over its life-cycle;

5. Empirical evaluation of whether informationally richer checklists actually reduce the documented problems, rather than relocate them.
6. Verification and oversight, so that an item marked complete, increasingly by an agent rather than a person, can be shown to correspond to work actually performed.

The arrival of agents does not only supply missing technology but also raises the stakes of our argument. In the introduction, we observed that our field has given considerable attention to a number of sophisticated coordination technologies, including BPMSs, Blockchain, and RPA, which comparatively few people use. At the same time, it has overlooked the checklists, which are nearly used by everyone. Agentic process management is the latest entry in that lineage, and a research agenda around it is forming quickly [4]. However, agents are no more a rival to the checklist. Since their behavior is autonomous and stochastic, the central concern of this emerging work is how to frame, bound, and verify what an agent does [10]. This is precisely the function of a checklist: an informational artifact that both represents a routine and prescribes the actions and decisions within it. The seven properties we proposed — representation, prescriptiveness, scope, abstraction, and audience, together with the type and behavioral relation of items — offer a vocabulary for specifying the scaffolds against which an agent acts and is checked. The community is reaching for ways to govern autonomous work. The checklist is among the oldest and most widely proven and has been in plain sight.

We think that this direction requires a scientific approach, which involves rigor, an explicit design orientation, close involvement with professionals from application domains, and ample space for experimentation and evaluation.

## 6 Conclusion

In this reflective paper, we argue that checklists are an important artifact to guide people through their work processes. Despite their demonstrated effectiveness across domains and time, their use suffers from problems that we relate to a lack of knowledge on how to properly design and maintain checklists. So far, we have not been able to mobilize substantial research efforts on this topic. However, we posit that there are technological advances that make a research agenda around this topic feasible and attractive. In particular, a mix of agent technology, LLMs, and process mining seem potent ingredients to address the core of the problems that professionals in various domains still suffer from.

With this paper, we also want to address a bigger issue, namely how a community selects the topics on which it works. The checklist is not a difficult artifact and we suspect that this is part of the reason why it has been overlooked. However, if the problems of the most widely used instrument for guiding work has no real place within the community that studies work processes, then this raises a question: Have we allowed the sophistication of our methods, rather than the importance of the problems, to set our research agenda? If this is true for checklists, could this be true for other topics as well?

It is not possible to provide meaningful answers to these questions on the basis of a single topic that did not make it into mainstream BPM or IS research. What we can do at this place is to plead for maintaining venues for exploratory work, room for "impact-first" rather than "puzzle-first" contributions, and organizing events where ideas that are *Forgotten*, *Overlooked* or *Rejected* get another chance to move into the spotlights.

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
