# OpenReview forum: "Why We Should Study Checklists: Reflections on an Overlooked Idea"
_bpm-conference.org/BPM/2026/Workshop/FOR-BPM — FOR-BPM_

### Official Review · Reviewer_LSSy · 2026-06-26
**A strong fit for the workshop wvvvith some opportunities for improvement**

**Rating:** 4

**Review:**

I enjoyed reading this one. It revisits the authors' 2017 paper on checklists as an informational artifact, and tells the honest story of what happened to it: a rejection at ECIS, an acceptance at HICSS, thirty-odd citations that mention it but never build on it, and a VICI proposal killed at the pre-proposal stage. Then it argues that process mining, LLMs, and agents make the old agenda worth reviving. For a workshop about overlooked and rejected ideas, you could hardly ask for a better fit, and the closing reflection on what the field chooses to study is exactly the kind of thing that gets a discussion going.

It's also clearly written and refreshingly candid. The authors don't hide the criticism they got. hey repeat it and admit a good deal of it.

My one real reservation is the thing the paper itself keeps circling. Both the reviewers and the grant committee said they couldn't see where the scientific challenge was, and the paper acknowledges this but never quite answers it. The revival section gives a six-point agenda, but it still doesn't say in plain words what the hard, general problem at the centre of a "science of checklists" actually is. This paper is the obvious place to finally settle that, and it stops just short.

A few smaller things hold it back too. The claim that the paper was cited but never built on rests on the authors' own informal reading of the citations; a quick systematic pass would make it stick. The argument that the field chases sophisticated technology over real impact is persuasive but anecdotal, and a little bibliometric evidence would go a long way. And the checklist still isn't cleanly separated from its obvious cousins, the to-do list, the SOP, the process model, the DMN table, which leaves the "isn't this trivial?" question half-open.

Strengths: an ideal fit for the venue, honest and self-aware, a genuinely provocative central point, and a timely link to agents and LLMs.

Weaknesses: it names the "where's the science?" problem without solving it, leans on anecdote in a couple of places, and doesn't fully pin down what a checklist is as opposed to its neighbours.

Minor: a couple of typos ("checklits" on pp. 2 and 6, a stray capital in "better training, Rather, they" in Sec. 2.1)

**Advancing Bpm Thinking:**

The single most useful move would be to finally say out loud what the science of checklists is. The material is already in the authors' own framework, so this is within reach. The cleanest way to show it is to take one of the twenty-one problems, say context-insensitivity, and work it all the way through: the problem, the property of the artifact that causes it, a hypothesis you could actually falsify, and how process mining or an LLM or an agent would test a fix. One worked example would do more than another page of argument.

It would also help to treat the seven properties as predictions rather than labels. The claim that certain properties cause certain problems is really a falsifiable theory, and saying so turns the framework from a taxonomy into something testable.

The idea would look far less trivial if it were connected to the literatures it sits next to but never cites: human factors and error theory, safety science, the IS work on representations, quality management and poka-yoke, clinical decision support. There's a real base there to build on.

On the agent argument, I would love to see it made concrete enough to test. Does an agent given an explicit checklist outperform an unstructured one on completeness and verifiability, and why is a checklist the right scaffold rather than the guardrails and tool-use schemas people are already building? And positioning the whole agenda as part of the Agentic BPM manifesto would give it the community home whose absence the paper itself diagnoses.

A few questions for the discussion:
- If you had to name the single deepest scientific question here, what is it?
- How do you tell a checklist apart from a to-do list, an SOP, a process model, or a DMN table?
- Why is the checklist the right scaffold for governing agents rather than the alternatives already in use?

---

### Official Review · Reviewer_r4kE · 2026-06-30
**Revisiting the idea of checklists as an artifact, making observations about the dynamics of research attention, and a case for current relevance**

**Rating:** 4

**Review:**

The paper revisits an idea that was raised a decade ago but was not uplifted then – checklists as an informational artifact. It describes the work that was published in the 2016 paper – findings, ideas, and research agenda, discusses possible reasons for the lack of uptake, and emphasizes the relevance of these ideas in the current technological context.

The paper is well written and definitely fits the workshop.

I agree with the observation that research aims at sophisticated technologies. But let’s put it this way: checklists are very simple means, commonly used by people without being backed by technology (or at least, not in a formalized way). The good thing about it – it is simple and it works (largely, despite issues that were raised).
In some ways, it can be embedded in technological solutions that have already been proposed over the years. Examples include milestones (e.g., in GSM) and preconditions in BPMS. These solutions could operationalize checklists, but lack the simplicity that makes them so prevalent. Any formalized technological implementation might suffer from the same… this may be another explanation for why this idea was abandoned. The other possible explanations that are given in the paper make sense and form an interesting reflection on research dynamics.

The proposal to use checklists in the current context of LLMs (to process them) and AI agents (for execution) makes a lot of sense. In this context the question that comes to mind is what should be the connection between checklists and goals (which agents should pursue). Would checklists be used for planning? Do they form a set of constraints, independent of a goal that guides the agent?

A minor comment: I wouldn’t say that blockchain has not been followed in practice.

**Advancing Bpm Thinking:**

I believe the paper does so in two ways:
1. A reflection on what makes an idea appealing for the research community (at a given context and point in time)
2. Re-raising the checklist idea and showing its relevance in the current AI-related context

These two can yield valuable discussions in the workshop and potentially gain the attention that was not there a dacade ago.

---

### Official Review · Reviewer_Xwc7 · 2026-07-02
**Everyone loves a good checklist but nobody wants to study it?**

**Rating:** 4

**Review:**

I really enjoyed reading this paper. Checklists are a basic and well established idea and the authors were right to point out the link with the BPM field in a prior (HICSS) publication. The paper is very reflective and explains well the initial concept, the impact (or lack thereof) since, and why now, given advanced in process mining and also AI technologies, is the right time to revive the idea. I think this would make for a great discussion in a BPM focused workshop.

In terms of the paper itself, the diagnosis section might benefit from some tightening. Explanations 2 and 3 are plausible. Explanation 1 is not an explanation at all for why an idea doesn't gain traction - everyone knows citations are not equivalent to uptake. I'm not sure explanation 4 is really an explanation for anything, given we don't know the quality of the applications (authors themselves note in section 5 that the proposal had room for improvement) and given we know that *most* grant applications get rejected (which doesn't cause the idea to die).

**Advancing Bpm Thinking:**

The paper reiterates six potential areas in which checklists could be studied in the context of BPM. There is also an opportunity here to link BPM to common time management methodologies. Getting Things Done is one example of those, that is hugely dependent on checklists. Some of the ideas proposed (e.g. living checklists) therefore have implications for the management field as well.